# Composable and Reusable Neural Surrogates to Predict System Response of Causal Model Components

**Ranjan Anantharaman** [1], **Anas Abdelrehim** [2], **Francesco Martinuzzi** [2,3,4], **Sharan Yalburgi** [2], **Elliot Saba** [2], **Keno Fischer** [2], **Glen Hertz** [2], **Pepijn de Vos** [2], **Chris Laughman** [5], **Yingbo Ma** [2], **Viral Shah** [2], **Alan Edelman** [1,2], **Chris Rackauckas** [2]

[1] Massachusetts Institute of Technology, [2] Julia Computing Inc., [3] Remote Sensing Centre for Earth System Research, Leipzig University, [4] Center for Scalable Data Analytics and Artificial Intelligence, Leipzig University, [5] Mitsubishi Electric Research Lab

## Abstract

Surrogate models, or machine learning based emulators of simulators, have been shown to be a powerful tool for accelerating simulations. However, capturing the system response of general nonlinear systems is still an open area of investigation. In this paper we propose a new surrogate architecture which is capable of capturing the input/output response of causal models to automatically replace large aspects of block model diagrams with neural-accelerated forms. We denote this technique the Nonlinear Response Continuous-Time Echo State Network (NR-CTESN) and describe a training mechanism for it to accurately predict the simulation response to exogenous inputs. We then describe a science-guided or physics-informed surrogate architecture based on Cellular Neural Networks to enable the NR-CTESN to accurately reproduce discontinuous output signals. We demonstrate this architecture on an inverter circuit and a Sky130 Digital to Analog Converter (DAC), showcasing a 9x and 300x acceleration of the respective simulations. These results showcase that the NR-CTESN can learn emulate the behavior of components within composable modeling frameworks and thus be reused in new applications without requiring retraining. Together this showcases a machine learning technique that can be used to generate nonlinear model order reductions of model components in SPICE simulators, Functional Markup Interface (FMI) representations of causal model components, and beyond.

*Keywords*: nonlinear system response, surrogate modeling, causal modeling, composable abstractions

## Introduction

Optimization of system designs commonly requires thousands to millions of expensive model simulations in order to identify to global or local minimums (Du et al. 2018; Yildiz, Abderazek, and Mirjalili 2020). To address this computational cost, reduced order and surrogate models (Willard et al. 2020) have become standard practice in many domains to accelerate simulations (Chatterjee, Chakraborty, and Chowdhury 2019; Han et al. 2017). Surrogates, and in particular physics-informed surrogates (White et al. 2019), are popular as stand-ins for full order simulations. They are also widely used in optimization (Stern, Song, and

Work 2017), uncertainty quantification (Tripathy and Bilionis 2018), and control design (Peitz and Dellnitz 2018).

Training surrogate models presents two difficulties. First, many systems used in engineering practice exhibit stiffness and have dynamics that are multi-scale in time (Wanner and Hairer 1996), which makes training accurate surrogates that capture all timescales a difficult task (Hadjinicolaou and Goussis 1998; Anantharaman et al. 2021b). Most data-driven methods for generating surrogates fail on this task without the use of problem-specific assumptions, variable transformations (Qian et al. 2020; Kramer and Willcox 2019) or specialized training procedures (Kim et al. 2021; Ji et al. 2021). Prior work demonstrated that implicitly-trained continuous time frameworks, such as the Continuous-Time Echo State Networks (CTESN), are viable surrogate architectures for accurately learning such multi-scale dynamics (Anantharaman et al. 2021b; Rackauckas et al. 2021).

A second challenge is the difficulty of developing surrogates which do not require retraining when used in new scenarios. For instance, in design optimization as well as optimal control, many surrogates directly approximate the objective function (Peitz and Dellnitz 2018; Marzat and Piet-Lahanier 2012; Wang, He, and Liu 2017), which necessitates retraining whenever the choice of objective function changes. Most other surrogate architectures reproduce the behavior of a full model (Anantharaman et al. 2021b; Raissi, Perdikaris, and Karniadakis 2019; Benner, Gugercin, and Willcox 2015). The downside this presents is that if the causal connections between components in a model are even slightly changed then the entire expensive training procedure must be repeated.

To alleviate this issue, we sought to develop surrogate architectures that would be suitable as components within composable modeling frameworks. These causal frameworks such as Simulink, ASPEN, and ModelingToolkit (Karris 2006; Luyben 2013; Ma et al. 2021), are block-diagram system modelers where mechanistic models are composed together. These types of simulation environments provide pre-built model components of commonly reused structures, such as HVAC models (Tian et al. 2017), buildings (Wetter 2011; Wetter et al. 2019), circuit components (Cellier, Clauß, and Urquía 2007), and more, to allow engineers to easily re-purpose quantums of mechanistic models towards different applications. If such component mod-

els could automatically be reduced to an accelerated form, then all future applications which would have used the library components could instead use the same neural surrogate without requiring retraining. It is this composable and reusable formulation that we wish to target.

Block models in such modeling systems are ordinary differential equations which allow for arbitrary input functions. This can be written as:

$$u' = \varphi(u, p, t, f(t)) \qquad (1)$$
$$y = \psi(u) \qquad (2)$$

where $u$ are the states of the component, $f$ is the input function, and $y$ are the output observables. In such a mechanistic system modeling context, systems are composed by defining the $f(t)$ by the output observables $y(t)$ of another block. Developing a surrogate architecture capable of being used in such a context thus boils down to developing continuous-time surrogates capable of capturing the system response of non-linear systems.

The problem of learning reduced system response models has been extensively studied in the context of control theory. A common case is where systems are linearized (Charlet, Lévine, and Marino 1989) about a certain operating point, after which linear model order reduction methods (Antoulas 2005; Benner, Gugercin, and Willcox 2015) or system identification methods (Ljung, Chen, and Mu 2020; Schoukens and Ljung 2019) are used to generate stand-ins for the full order model. Then, the literature on designing controllers for linear systems can then be drawn upon (Skogestad and Postlethwaite 2007). The limitation of this approach is that beyond a certain operating region, this reduced order form's system response may no longer be representative of the original system response. One method that has been used to capture nonlinear system response is the Volterra series (Cheng et al. 2017), and have been used to control multi degree of freedom systems (Feijoo et al. 2010) but the method requires the system to be time-invariant. In this work, we present a fully nonlinear machine learning based method to capture the system response of arbitrary nonlinear systems in a continuous-time architecture and show its ability to accurately accelerate causal models.

## Nonlinear Response Continuous-Time Echo State Networks (NR-CTESN)

This section first summarizes the CTESN surrogate method and then describes the Nonlinear Response CTESN extension. CTESNs are a continuous-time analogue of Echo State Networks, which are a form of reservoir computing (Grigoryeva and Ortega 2018; Lukoševičius 2012). This surrogate consists of two parts: a reservoir Ordinary Differential Equation (ODE), which is cheap to simulate by design, and a projection operator from this reservoir to the reference system. While the reservoir ODE is designed by the user based on various considerations, the projection operator is trained using a linear least-squares solve. A typical formulation for the CTESN is the following:

$$r' = g(Ar + W_{hyb}x(p^*, t)) \qquad (3)$$
$$x(p, t) = W_{out}r(t) \qquad (4)$$

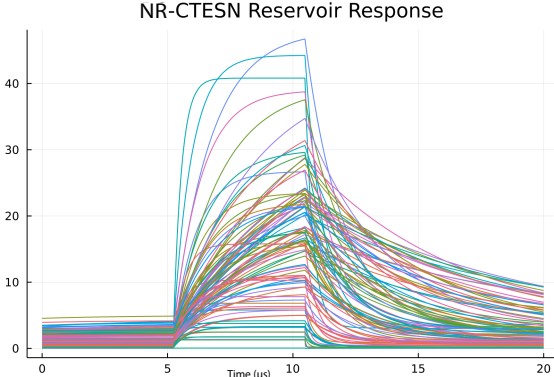

Figure 1: **Rich response dynamics of a domain-specific reservoir used in the NR-CTESN**, used to generate the surrogate of the inverter. Each line represents an output from every cell in the cellular neural network.

where Equation 3 is an example of a simple, non-parametric reservoir ordinary differential equation, written as a neural ordinary differential equation(Chen et al. 2018). $A$ is referred to as the weight matrix in the literature (Kawai, Park, and Asada 2019), $W_{hyb}x(p^*, t)$ is a "hybrid" term used to drive the reservoir, and $x(p^*, t)$ is a candidate solution at some point in the chosen parameter space. $g$ is an activation function, which, in addition to the weight matrix $A$, is chosen to control the behaviour of the full derivative term.

The CTESN is conventionally trained as follows: first, a parameter space $P$ is chosen, which is a cross product of ranges of the system parameters. This space is then sampled, yielding $\{p_1, \ldots, p_n\} \in P$. The system is simulated at each of these parameters, yielding a training set of time series. Equation 4 refers to the second step in the training. Projections $\{W_{out}^{p_1}, \ldots, W_{out}^{p_n}\}$ are computed from the simulated reservoir $r$ to each time series in the training set, using a QR decomposition or the singular value decomposition. Finally, an interpolating function $p \rightarrow W_{out}(p)$ is then fit. Prediction from the CTESN now follows three steps: simulation of the reservoir, constructing the projection, and then matrix-vector multiplication, as shown below.

$$\hat{x}(\hat{p}, t) = W_{out}(\hat{p})r(t) \qquad (5)$$

The CTESN has been trained to produce surrogate models of Heating, Ventilation and Air Conditioning (HVAC) systems (Rackauckas et al. 2021) and quantitative systems pharmacology models (Anantharaman et al. 2021a).

There are several important considerations whilst training the CTESN. The reservoir should be always be chosen strategically. In particular, the time series from this reservoir should match key characteristics of the output time series, while being cheap to simulate. The above basic formulation may not be amenable to train systems that exhibit complex behaviour. For instance, this reservoir is largely continuous, and may not accurately capture discontinuities. If the output time series contains a discontinuity at a point in time, the reservoir should also contain it. In the next section, we shall examine a domain specific choice of reservoir which can accurately handle semi-discontinuous components seen in circuit simulations.

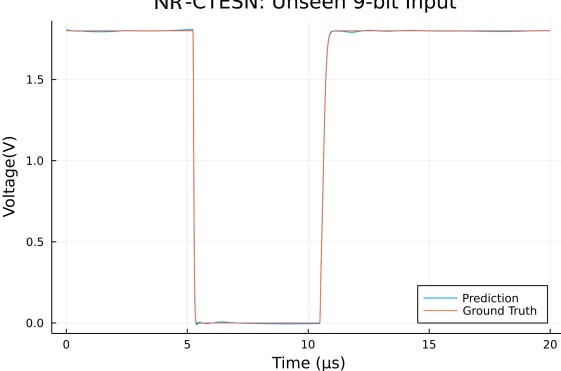

Figure 2: **Prediction of surrogate to unseen test input sequence**. The red line refers to the ground truth output from the inverter and blue line is the prediction. The surrogate is able to predict the transition from high to low at the correct times and is able to match the reference output.

The NR-CTESN incorporates external forcing into the formulation and training. While external forcing can also be handled by a vanilla CTESN by simply appending the function parameters to the list of physical parameters, the resulting surrogate does not distinguish between these two classes of parameters. This treatment is incorrect because the forcing parameters affect system dynamics differently than physical parameters. The NR-CTESN thus treats these parameters differently by incorporating the forcing term into the reservoir. Using this modification, discontinuities can also be incorporated into the reservoir formulation, provided they are known in advance of training.

While training the NR-CTESN, we do not simply sample a larger Cartesian space, like in the vanilla CTESN. Instead, we sample the cross product of two Cartesian spaces, corresponding to the physical parameters and forcing parameters. This allows the NR-CTESN to train over many different combinations of physical and forcing parameters, thus providing a more accurate and robust surrogate. The following section describes this training procedure in more detail.

### Training with External Forcing

In this section, we describe a modification to the structure and the training procedure of the CTESN Now we present a variant of the CTESN formulation that incorporates external forcing, which can then be used to capture system response to external forcing or internal disturbance. This training procedure not only trains on a bounded parameter space $P$ like the conventional CTESN, but also on a space of external forcing functions.

Consider a bounded space of forcing functions $F$, such as a Fourier series or polynomials with a finite number of terms and a predefined range of coefficients. In other words, we mean to consider a bounded space of coefficients that describe a bounded space of functions. Let forcing functions $\{f_1, \ldots, f_n\}$ be sampled from this space. To incorporate forcing functions, Equation 3 may thus be modified as follows:

$$r' = g(Ar + W_{hyb}x(p^*, t) + W_f f(t)) \qquad (6)$$

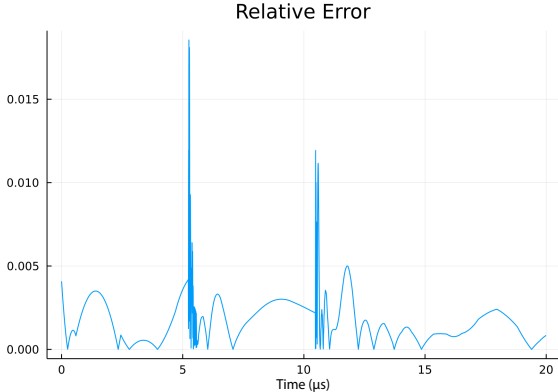

Figure 3: **Relative error of surrogate prediction at test input sequence.** The error is highest at the points of discontinuity: the places where the square signal transition from 0 to 1 and vice versa.

where $W_f$ is constant and $f(t)$ is the additional forcing term. The reservoir $r$ from Equation 6 is excited using each forcing function, resulting in a collection of reservoir response time series $\{r_1, \ldots r_n\}$. We also excite the original system with the same forcing functions, resulting in the time series collection $\{d_1, \ldots d_n\}$. We can now solve the least squares problem

$$\arg\min_{W_{out}} \sum_i (W_{out}r_i - d_i)^2$$

To solve this numerically, we can concatenate all the system responses like in Equation 7, and solve the following directly with a QR decomposition or the singular value decomposition:

$$W_{out}[r_1|r_2|\ldots|r_n] = [d_1|d_2|\ldots|d_n] \qquad (7)$$

We dub the resultant surrogate the NR-CTESN. This idea composes naturally with the conventional CTESN training to learn a parametric surrogate. Namely, we can sample several sets of parameters at pre-defined parameter space $P$, compute several projections from the reservoir system responses to the original system responses, and then learn the interpolating object $W_{out}(p)$. This procedure is summarized in Algorithm 1. We also note that system response to internal disturbances and events can be learnt in the same way. Once the event in question is parametrized, a space of event parameters can be bounded and sampled, after which a projection can be fit in exactly the same fashion as above.

### Circuit Simulation Applications: Inverter and Digital to Analog Converter (DAC)

In this section, we shall discuss how the Nonlinear Response CTESN is used to generate a surrogate of an inverter circuit. The inverter is designed using two Berkeley Short-channel IGFET Mode-4 (BSIM4) transistor models (Dunga et al. 2006) in CMOS inverter configuration. The input to this model is a 9-bit digital input with a fixed bit-width and the output is the flipped bitstream.

The first step in training a NR-CTESN is to design the reservoir. In general, the reservoir should exhibit a rich set

Algorithm 1: Training the Nonlinear Response CTESN

**Input**: parameter space P, space of forcing functions F, pre-designed reservoir $r$

**Output**:

1: Sample $\{p_1, \ldots p_n\} \in P$, $\{f_1, \ldots f_n\} \in F$
2: **for** p in $\{p_1, \ldots p_n\}$ **do**
3:   Compute reservoir responses $\{r_1, \ldots r_n\}$ and system responses $\{d_1, \ldots d_n\}$ using $\{f_1, \ldots f_n\}$.
4:   Fit projection matrix $W_{out}$ at parameter $p$ using Eq. 7
5: **end for**
6: Fit interpolator $\psi : p \to W_{out}$
7: **return** $\psi, r$

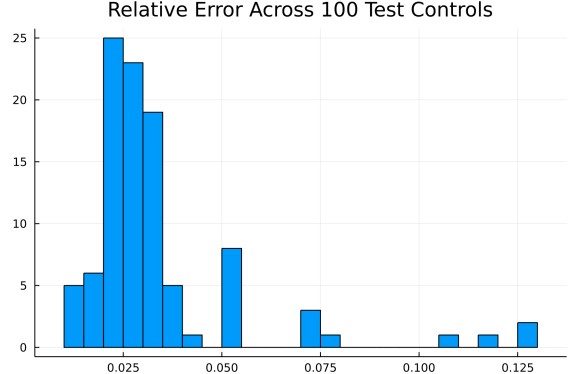

Figure 4: **Histogram of test errors**. The X axis denotes the relative error and Y axis denotes the number of test samples with the same prediction error. The majority of relative errors for the test dataset is less than 0.05.

of dynamics when excited by the digital inputs. The inverter is a mixed-signal circuit (Gielen and Rutenbar 2000), and can output both analog and digital signals depending on its physical parameters. The reservoir must also follow this behaviour.

This was achieved by use of a domain-specific reservoir in the form of a Cellular Neural Network (Chua and Yang 1988a), which consists of a grid of cells. Each cell is a circuit with an RC element, with additional controlled current sources denoting coupling with its neighbors. The state equations are as follows:

$$Cv'_{ij} = -\frac{1}{R_x}v_{ij} + \sum_{C(k,l)\in N_r(i,j)} A(i,j;k,l)v_{ykl}$$
$$+ \sum_{C(k,l)\in N_r(i,j)} B(i,j;k,l)v_{ukl} \quad (8)$$

$$v_{yij} = g(v_{ij}) \quad (9)$$

where $v_{ij}$ is the state of each cell $C(i,j)$, the resistances $R_x$ and capacitances $C$ in each cell are randomly initialized, which allows them to create a range of responses. The output from each cell is the state of the cell $v_{ij}$ filtered by an activation function $g$, chosen to be $\tanh$ in this example. Each cell also interacts with its neighbors via two template variables $A$ and $B$, which dictate the weighted contribution from the

neighbors' outputs $v_{ykl}$ and inputs $v_{ukl}$ respectively (Chua and Yang 1988b; Hunt et al. 1992). The size of the neighborhood $N_r(i,j)$ controls the size of the two template variables. This coupling in the system produces a rich response shown in Figure 1. For more details on the implementation, the reader is referred to the basic cellular neural network formulation in (Chua and Yang 1988b).

Since the input is 9-bit digital, there exist only $2^9 = 512$ possible discontinuous input signals to this system. The location of the discontinuity is controlled by inverter's system parameters, and the surrogate is trained on multiple sets of system parameters sampled using Latin Hypercube sampling. Additionally, it is trained with 100 inputs sampled from the possible 512. A $10 \times 10$ cellular network is randomly initialized and excited with the training inputs, and a neighborhood of $3 \times 3$ was chosen. The simulation time under consideration for training is 20 micro-seconds. A least squares projection is then calculated between the reservoir system responses and the system responses of the original system. Figure 2 shows a prediction plot of the surrogate at a test input while Figure 3 shows the relative error, which is less than 2% over the whole time series. Additionally, a histogram of relative errors was plotted in Figure 4.

On the inverter example of 44 equations a 9x acceleration of the simulation was achieved by the NR-CTESN. However, as demonstrated in previous work the CTESN architecture's acceleration increases as the size of the approximated system increases (Anantharaman et al. 2021b,a). This is true of the NR-CTESN architecture as well. To demonstrate this, we trained a surrogate using the same architecture on a Sky130 Digital to Analog Converter (DAC) simulated by Ngspice (Nenzi and Vogt 2011). This 1,200 equation system saw similar accuracy results but achieved a 274x acceleration, changing the simulation time from 7.3 hours to 1.6 minutes.

## Discussion & Conclusion

This paper presents a method to learn the response of arbitrary systems to external forcing or internal events. The method presented is generalizable in that it is data-driven and places no constraints on the nature of the system being approximated. Science-guided or physics-informed priors are shown to be incorporated by the choice of reservoir. Yet, the technique is, in general, agnostic to the system being learned. The resulting surrogate is capable of generating nonlinear model order reductions of casual modeling blocks that match the reusable component architecture. This allows the NR-CTESN to automatically accelerate models from these types of modeling frameworks. When applied to the Functional Markup Interface (FMI) standard for causal models (Blochwitz et al. 2011), a widely used binary form describing models in a way that matches Equation 1, the NR-CTESN can thus be used as an FMU-accelerator, taking in FMU binary descriptions of causal components and generating new FMU binaries which reproduce the behavior at a fraction of the cost. Given the hundreds of widely used tools throughout industry engineering which use this standard[1],

---

[1]https://fmi-standard.org/tools

this technique has the potential to make a large impact on the modeling industry.

In future work, we plan to use this surrogate in order to design controls. We anticipate that the ability of the surrogate to respond to control signals at a fraction of the computational cost can accelerate design significantly. In addition, we note that the original discrete-time echo state networks have been demonstrated to be universal adaptive filters (Grigoryeva and Ortega 2018), suggesting a potential universality for the NR-CTESN. Follow up work should investigate this property or suggest variants to the NR-CTESN to correct for this ability. Finally, we note that this technique requires knowing the input states of the system and thus cannot capture the fully acasual modeling space of many system modelers such as Modelica (Mattsson, Elmqvist, and Otter 1998), Dymola (Brück et al. 2002), or SimScape (Miller and Wendlandt 2010). Extensions to the technique which cover the acausal space would expand its applicability.

## Acknowledgments

This material is based upon work supported by the National Science Foundation under grant no. OAC-1835443, grant no. SII-2029670, grant no. ECCS-2029670, grant no. OAC-2103804, and grant no. PHY-2021825. We also gratefully acknowledge the U.S. Agency for International Development through Penn State for grant no. S002283-USAID. The information, data, or work presented herein was funded in part by the Advanced Research Projects Agency-Energy (ARPA-E), U.S. Department of Energy, under Award Number DE-AR0001211 and DE-AR0001222. We also gratefully acknowledge the U.S. Agency for International Development through Penn State for grant no. S002283-USAID. The views and opinions of authors expressed herein do not necessarily state or reflect those of the United States Government or any agency thereof. This material was supported by The Research Council of Norway and Equinor ASA through Research Council project "308817 - Digital wells for optimal production and drainage". Research was sponsored by the United States Air Force Research Laboratory and the United States Air Force Artificial Intelligence Accelerator and was accomplished under Cooperative Agreement Number FA8750-19-2-1000. The views and conclusions contained in this document are those of the authors and should not be interpreted as representing the official policies, either expressed or implied, of the United States Air Force or the U.S. Government. The U.S. Government is authorized to reproduce and distribute reprints for Government purposes notwithstanding any copyright notation herein.

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
