# OpenReview forum: "Composable and Reusable Neural Surrogates to Predict System Response of Causal Model Components"
_AAAI.org/2022/Workshop/ADAM — AAAI 2022 Workshop ADAM_

### Official Review · Reviewer_xBsv · 2021-11-29
**Paper with a strong motivation, but results section could be strengthened, and existing one figure in the result presented is incomplete.**

**Rating:** 6
**Confidence:** 2

**Review:**

This paper proposes a new type of surrogate model non-linear called Continous Time Echo State Network (NR-CTESN) that is built upon a previously proposed Continous Time Echo State Network (CTESN) model. The CTESN model is interesting because it has previously been shown to be able to capture systems with multi-scale dynamics. The NR-CTESN model improves upon the CTESN model by adding the ability to incorporate external forcing functions to the generic formulation of CTESN. Experiments were shown on two case studies for an inverter and a digital-to-analog converter, which shows that the proposed surrogate models can accurately emulate these systems and accelerate the simulations significantly. The introduction and formulation of this paper were written well with a motivation that is of great significance to real applications. However, the results section can be improved upon to make the results more convincing.

Specifically, in Figure 2, the prediction line is not seen, and the red and blue line referenced in the caption is also not seen in the figure.

Additional results for the inverter would have made the results section stronger.

Also, i(t) is not defined in the second page left column when describing Equations 1 and 2

---

### Official Review · Reviewer_dvgV · 2021-12-01
**Marginally novel paper with interesting applications**

**Rating:** 5
**Confidence:** 3

**Review:**

The paper extends Continuous Time Echo State Networks (CTESN) for surrogating systems with nonlinear reponses. The method presented in the paper changes the CTESN formulation by adding a forcing term (in terms of bounded basis functions) allowing CTESN type networks to capture inherent discontinuities or external effects. This allows CTESN type models to now be used to surrogate models with nonlinear responses. In addition the authors propose such models to be composable, thus allowing for modelling complex systems with multiple components. They show an application of the same with an 9-Bit inverter.

Pros:
1. The paper shows some interesting applications of the modified CTESN surrogates.
2. NR-CTESN performs well for the presented application.

Cons:
1. NR-CTESN seems to be an obvious modification of CTESN. The authors do not clearly point out why such a modification is non-trivial.
2. In my understanding,  CTESN can also solve the same problem, albeit with careful design, which NR-CTESN helps in avoiding. Could the authors elaborate the difference in terms of the provided inverter example? From observation, it appears that the inverter model also requires careful design.
3. Figure 2 is missing the predicted response.
4. Additionally, the paper does not present any quanititative evidence of the NR-CTESNs for the DAC application, which probably is more interesting.
5. Finally, the paper could be better motivated by presenting if and how the CTESN model fails for the given examples while NR-CTESN succeeds.

In summary, the paper does not present a clear motivation of why NR-CTESN is better than CTESN, and has some glaring errors. I recommend a borderline reject, unless the authors fix such errors in the final draft.